# Wireless Sensor Network for Ignitions Detection: An IoT approach

**Thadeu Brito** [1,†] , **Ana I. Pereira** [1,2,†] , **José Lima** [1,3,†] **and António Valente** [3,4,*,†]

1   Research Centre in Digitalization and Intelligent Robotics (CeDRI), Instituto Politécnico de Bragança, Campus de Santa Apolónia, 5300-253 Bragança, Portugal; brito@ipb.pt (T.B.); apereira@ipb.pt (A.I.P.); jllima@ipb.pt (J.L.)
2   Algoritmi Research Centre, University of Minho, Campus Azurém, 4800-058 Guimarães, Portugal
3   INESC TEC—INESC Technology and Science, 4200-465 Porto, Portugal
4   Engineering Department, School of Sciences and Technology, UTAD, 5000-801 Vila Real, Portugal
*   Correspondence: avalente@utad.pt
†   These authors contributed equally to this work.

**Abstract:** Wireless Sensor Networks (WSN) can be used to acquire environmental variables useful for decision-making, such as agriculture and forestry. Installing a WSN on the forest will allow the acquisition of ecological variables of high importance on risk analysis and fire detection. The presented paper addresses two types of WSN developed modules that can be used on the forest to detect fire ignitions using LoRaWAN to establish the communication between the nodes and a central system. The collaboration between these modules generate a heterogeneous WSN; for this reason, both are designed to complement each other. The first module, the HTW, has sensors that acquire data on a wide scale in the target region, such as air temperature and humidity, solar radiation, barometric pressure, among others (can be expanded). The second, the 5FTH, has a set of sensors with point data acquisition, such as flame ignition, humidity, and temperature. To test HTW and 5FTH, a LoRaWAN communication based on the Lorix One gateway is used, demonstrating the acquisition and transmission of forest data (simulation and real cases). Even in internal or external environments, these results allow validating the developed modules. Therefore, they can assist authorities in fighting wildfire and forest surveillance systems in decision-making.

**Keywords:** WSN; IoT; fire detection; LoRaWAN; wildfires

## 1. Introduction

Wireless Sensors Network (WSN) is actually used to build decision support systems that can be used in several fields. One of the most interesting fields having an increasing need for decision support systems is agriculture and forestry. These areas have been neglected by technology, but today this trend is changing. Smart sensors can be useful in acquiring data from nature, and acquiring useful data for decision-making. Examples are precision farming [1] and fire detection [2]. Concerning this last topic, many measures have been taken to identify fires and detect them as soon as possible.

The Forest Alert Monitoring System (SAFe) project proposes to develop and install a set of innovative operations to minimize the alert time of forest fire ignitions. This project will contribute to the actual surveillance systems (such as cameras and surveillance towers), enhancing the firefighters and civil protection with more details and real-time information. The base of this project is the acquisition and communication modules, that will be spread in the forest, and will gather information about several relevant data for efficient characterization of existing forest conditions (fire ignition detection and danger index, as examples). This information will be processed using Artificial

Intelligent methodology that will trigger the fire occurrence and also warnings of dangerous situations. This methodology will be presented in a future paper. The present work is to address the modules, its characteristics and the hardware developed to acquired the forest data. There are requirements that push the development of different module approaches. These requirements, such as the measurement types, modules sizes, supply modes (photovoltaic model) and price due to theft conditions were decisive points for choosing the development of two types of modules, both communicating through LoRaWAN. The first one, named HTW, will handle several sensors and was designed to be expandable since it owns SDI-12 communication protocol and I2C. The second one, named 5FTH, will be a cheap and straightforward approach that will acquire five flame sensors, temperature, and air humidity. In this paper, both models will be addressed where 5FTH module is in testing stage whereas the HTW module is already in forest environment acquiring data.

This work will detail the development of both modules and present the results of acquisition, transmission, and visualization of the data. The final prototype composed by both modules and the transmission system is validated and can be further used in forest to detect fire ignitions. The presented work will contribute to the actual surveillance systems, given to the firefighters and civil protection more useful data and real time information about the forest.

This paper is organized as follows. After an introduction in Section 1, related work about Wireless Sensors Network and fire detection is presented in Section 2. In Section 3 SAFe system architecture is described. The characterization of the two wireless sensor modules is done on Section 4 and the obtained results are presented in Section 5. Finally, Section 6 concludes the paper and points some future work direction.

## 2. Related Work

Due to the development of new technologies within the scope of the Fourth Industrial Revolution, the forestry sector can be digitized in terms of resolving issues such as rural fires [3]. Integrating several systems, making them collaborative is a possible solution to fighting wildfires through an operating system to alert events based on these new approaches. In this sense, all the mentioned components will transform the nature reserves, making them become Forests 4.0 [4]. Regarding these topics, the related work section is divided into Internet of Things (IoT) platforms and fire detection subsections. Both discuss briefly published matter technically related to the present work.

### 2.1. WSN Platforms

WSN is an area that has been used to monitor remote variables. Environment acquisition can use WSN to get data from regions of difficult access, such as habitat monitoring [5]. WSN is composed by a large collection of sensors that can be used to gather data and send them by an IoT system. In fact, WSN is a technology used within an IoT system. There are several platforms available to support IoT, such as LoRaWAN, Sigfox, Ingenu, Weightless-N and NB-IoT. Sigfox is a service provider for IoT and LP-WAN a network operator that commercializes its own IoT solution as Network and as a Service (NaaS) in around 57 countries. On the other hand, LoRa (Long Range) is a technology that was developed by the startup Cycleo in 2009. Three years later, Cycleo was purchased by Semtech, a company from the USA. In 2015, LoRa was standardized by LoRa-Alliance that makes it an open software and IoT platform hardware. Furthermore, LoRa is deployed in around 100 countries. In LoRa's business model, users can deploy theirs own infrastructure.

LoRaWAN, Sigfox and Ingenu are analyzed in terms of efficiency, effectiveness, and architectural design, especially for smart city applications, in Marco Centenaro et al. work [6]. They also present practical results of experiments and deployments with IoT networks based on LoRaWAN, developed in the city of Padova, Italy. Low power consumption is an important topic in this area. Raza et al. [7] presents the most essential Low Power Wide Area (LPWA) platforms and technologies available in the market. Sinha et al. [8] and Mekki et al. [9] analyze and compare NBIoT and LoRaWAN platforms based on licensed and unlicensed bands, respectively. The platforms are compared, showing the

technical differences between Sigfox, LoRaWAN and NB-IoT in terms of IoT success factors such as quality of service, coverage, range, battery life, latency, payload length, scalability, deployment, and cost. The result of the comparison shows that Sigfox and LoRaWAN have advantages in terms of battery lifetime, capacity and cost. NB-IoT offers benefits in terms of latency and quality of service. In short, besides LoRaWAN having its advantages such as being licence free, we can manage the network, it is a bidirectional Low Power Wide Area Network and the other networks do not have coverage; LoRaWAN will be the one used in this work.

*2.2. Fire Detection*

Forest fires are uncontrolled fires occurring in wild areas and cause significant damage to natural and human resources. In fact, the faster the firefighters arrive at the fire localization, the easier is to control the fire [10,11], so it is a huge importance to detect the first ignition to alert the authorities. The first fire detection method was based on image processing [12]. It is a very interesting method in small areas, such as commercial areas, power plants, or forests that are not dense. However, in huge regions or dense forests, if a fire ignition happens, the fire identification will occur too late. Thus, other fire detection approaches, such as using a WSN, have been addressed for a long time. In forests the use of WSN has been studied by several authors, for example [2,13,14]. Some authors use the temperature, relative humidity, and barometric pressure values to detect the flame before the area has being scorched, since the temperature increase, and barometric pressure and humidity decrease as the fire evolves [15].

Moreover, a sensor fusion technique to enhance the performance of forest fire detection and location estimation is also presented in [16]. Fusion on light and temperature measurements can also be done to perform the fire detection [17]. Once direct sunbeam on the light sensor can be confused with the fire, Penha et al. propose and compare two methods to outwit this problem, the Threshold and the Dempster-Shafer Methods. Another significant input established to the fire model is the Fire Weather Index (FWI) [18]. It can be used to define the data rate transmission between modules and the central system. Having the previous authors' concepts, the present paper addresses a step forward in fire detection methodology using new sensors and approaches that speed-up the forest ignitions identification.

## 3. System Architecture

As already mentioned, SAFe aims to install a set of innovative activities to be developed in regions with higher potential for fire ignitions. Therefore, SAFe is implemented following an integrating strategy with some essential tools for its purpose. The grouping of these tools is illustrated in the Figure 1, where a target monitoring region is defined and will benefit from these proposed sets of components and applications.

The proposed system is based on four fundamental elements: the monitoring region, the sensors modules set, the communication system and the control center. The union between the elements coupled with an artificial intelligence-based system will enable an efficient and intelligent data analysis. The data analysis will promote the creation of hazard alerts, alerting rescue and combat teams (such as fire brigades, civil protection or town hall). These alerts will be parameterized and presented in a personalized way, according to each organization involved. Thereupon, the project can be expressed through eight categories that work collaboratively, identified by Figure 1; they are:

- The monitoring region (represented by ①) is the region where the Wireless Sensor Modules set will be placed. In this study case the monitoring region is placed in the Trás-os-Montes area;
- The Wireless Sensor Modules set (represented by ②) are responsible for the data acquisition at the forest in real-time. The coordinates of each Sensor Module is calculated through an optimization procedure that considers the hazard fire in each coordinate;

- The LoRaWAN Gateway (③) receives data from each sensor module and then forwards the data through a 4G/LTE link (or by Ethernet where available) to a server (represented by ④);
- The Control Center (represented by ⑥) analyzes all information and sends alerts for hazardous situations or forest fire ignitions to the surveillance agent in the region. This control center has a server (represented by ⑤) that store all collected data, and perform artificial intelligence procedures to correlate data from sensor modules with external data (represented by ⑦); such as local scale real-time fire hazard indexes, availability fuel content, weather data and moisture content of the vegetation.

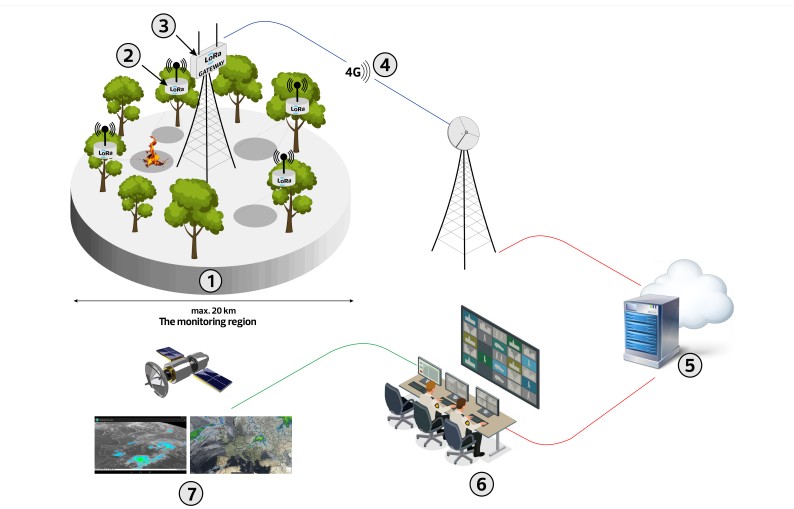

**Figure 1.** Illustration of Forest Alert Monitoring System (SAFe) system architecture [19]. ① The monitoring region, ② Wireless sensor module, ③ LoRaWAN gateway, ④ 4G/LTE link, ⑤ Server, ⑥ Control center, and ⑦ Detection support system.

Due to the complexity in describing and developing each of these items, this work will only focus on the Wireless Sensor Network and the proposed communication infrastructure. In this way, it is presented a discussion about different strategies on the WSN system that combines the sensors modules with the communication approach. Thus, to integrate the network is implemented two different types of sensors modules: HTW and 5FTH modules. The first one, HTW module, can be expandable since it owns SDI-12 communication protocol and I2C (changeable). The last one, 5FTH module, is based on low-cost price, having five flame sensors, temperature and air humidity (unchangeable). Both use LoRaWAN communication to send data to the control center. All these concepts are described, respectively, in the next section.

## 4. Wireless Sensor Modules

As we are in a new paradigm of IoT, each "thing" will have to own special characteristics such as being of small dimensions, being self-sufficient in terms of energy, being of low-cost (a considerable density of sensors will always be necessary), and essentially, having only one sensor (or set of sensors) for a given measurement. Therefore, two types of modules were developed for sensor nodes: one for connecting generic external sensors and another, more specific, and with sensors already defined and implemented in the module. The generic module, HTW, allows connection to digital sensors with communication through the SDI-12 protocol, as are the cases of sensors used in agriculture and which will be necessary for the system. Examples of this are ultrasonic wind speed sensors (ATMOS22 from Meter Group [20]), all-in-one weather station (ATMOS41 from Meter Group [21]), and sensors for the soil (5TE from Meter Group [22] and Watermark by Irrometer [23]). As already mentioned, two modules were developed, according to the local requirements. These modules will be presented in the next subsections.

### 4.1. Sensor Module–HTW

The HTW module, depicted in Figure 2, is composed of the processing module, LoRaWAN communication and energy management with solar charging, it also has inputs compatible with some sensors to be used. In the developed system, standard sensors for agriculture can be used, which normally have SDI-12 communication [24], temperature, humidity and atmospheric pressure sensors that, in most cases, communicate by I2C (e.g., BME680 from Bosch [25]). Besides, it must have analog inputs (ADC) for, among others, can also have infrared flame detectors.

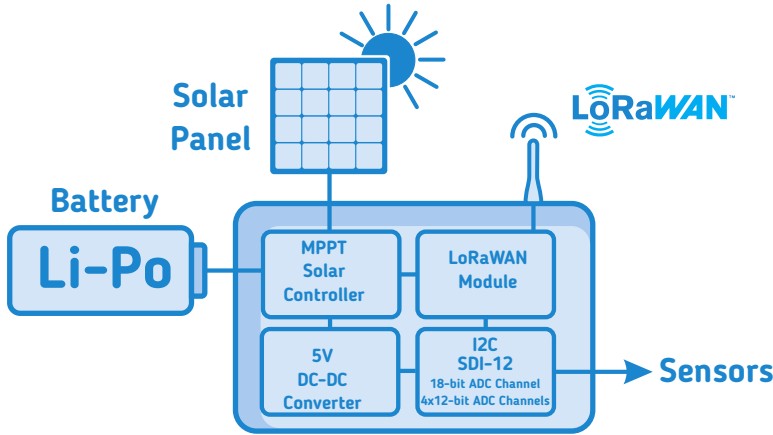

**Figure 2.** HTW mote overview.

With the view to contemplate all the characteristics presented, the HTW module has a CubeCell board [26] (with built-in microcontroller, LoRaWAN RF module, and solar battery charger for one cell LiPo battery), a single-channel 18-bit ADC, Microchip's MCP3421, and a 12-bit, 4-channel ADC, MAX11613 by Maxim. The CubeCell board has a UART, but it is busy for debugging and programming the module itself. Thus, to implement the SDI-12 communication protocol, it was also necessary to add a UART module by I2C (SC16IS740 from NXP). Figure 3 shows a simplified scheme of the developed system.

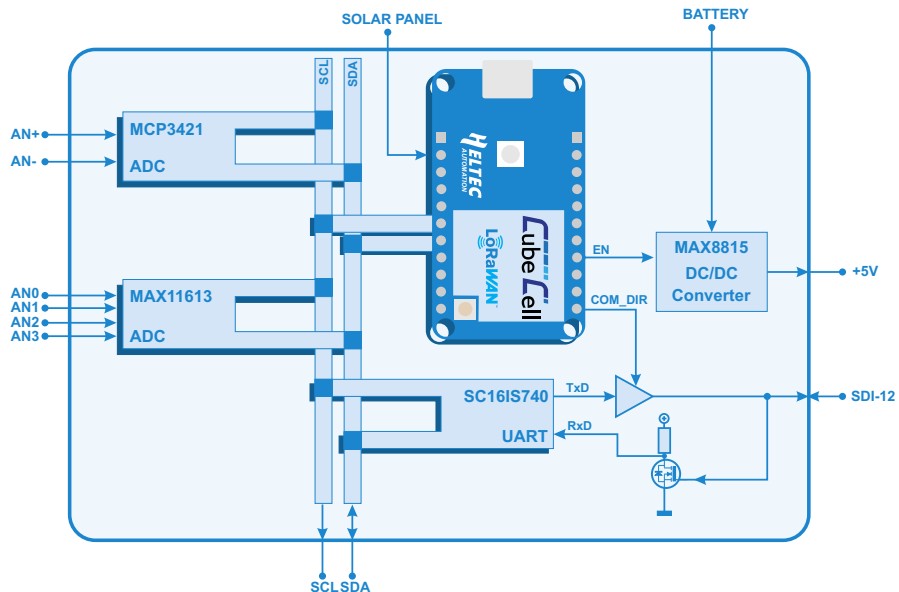

**Figure 3.** PCB mote simplified scheme with CubeCell as core.

In this regard, it is possible to carry out the development of the prototype HTW module demonstrated by the process sequence in Figure 4. In which Figure 4a shows only the PCB with

all the components, where you can see Heltec's CubeCell module. This process was necessary to verify basic functioning tests, that is, if not problems in the integration between the DC/DC converter, CubeCell, the UART and ADC modules. Then, in Figure 4b, a case is made using 3D printing to protect the PCB from weather events. Finally, the module completion with the addition of the Turnigy 2000 mA (1S 1C) LiPo battery and the solar panel for charging. The final box module has the following dimensions: 85 mm × 65 mm × 35 mm.

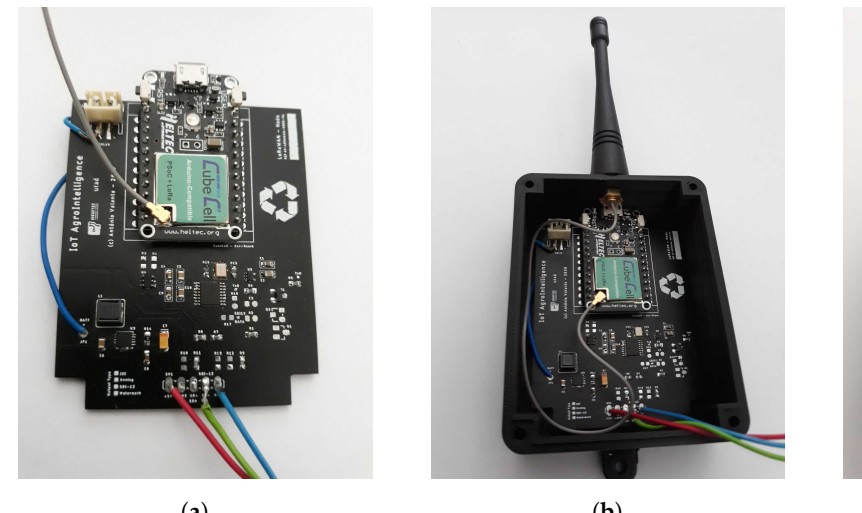
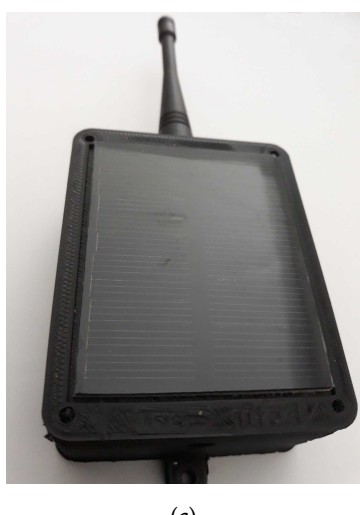

(**a**)　　　　　　　　　　　　　　　　(**b**)　　　　　　　　　　　　　　　　(**c**)

**Figure 4.** The process sequences to develop HTW module. (**a**) The developed PCB with components; (**b**) 3D printed case with PCB assembled; (**c**) HTW module solar panel.

*4.2. Sensor Module–5FTH*

This module was developed keeping in mind several characteristics that make it useful for forest places when a low-cost is required. Moreover, it could be placed in situations where theft or vandalism can happen. This module is also prepared to perform several measures, as will be presented in the next section. Figure 5 depicts the fundamentals components that were considered during the development. In which Figure 5a illustrates the main architecture of this developed board. Based on this, this board is powered by a 18,650 lithium-ion cell battery that will supply all the system. A converter is used to supply the entire components (except the microcontroller) at 3.3 V. By this way, a buck-boost converter guarantees the supply voltage of 3.3 V regardless of battery voltage (between 2.5 V and the maximum the battery when fully charged). This converter (presented in Figure 5b) has an enable pin that is being used to shut down all the board components (except the microcontroller). The sleep function of the microcontroller was activated to save energy consumption since this module will be placed with no grid connection. The microcontroller used for this module is the ATMega328P, a well-known device due to the Arduino open-source hardware and software project [27].

This microcontroller reads the environment temperature and humidity through the DHT11 component [28]. Other temperature and humidity sensors could be used due to the microcontroller flexibility, but DHT11 was chosen to be implemented in the prototype because to the low-cost, low installation complexity, and also because it has a relatively small measurement tolerance (±5% humidity and ±2° temperature). There are five analog inputs that are used to acquire the flame sensors values at a sample period of about 60 s that are transmitted to the upper level (the transmission frame rate will depend on the number of sensors and the risk analysis). If a flame is detected, the transmission event is triggered. Furthermore, this PCB owns the programming pins so that the firmware can be updated without necessities of pick-off the ATMega328P from the board. The LoRa module used for this approach is the RFM96, as presented in Figure 5c, that is a low power wireless transceiver module for LoRa transmission. It is connected to the microcontroller via an SPI port and supplied by the DC/DC converter (when the enable pin is activated).

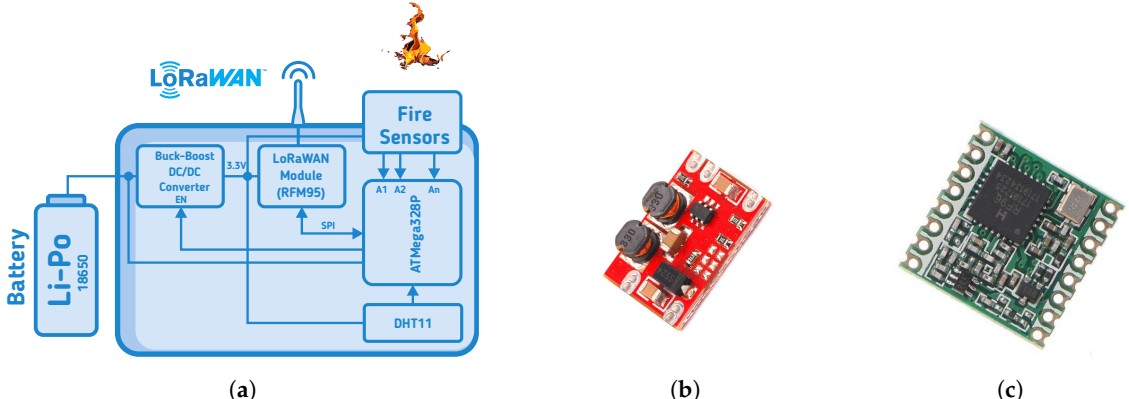

|  (a) | (b) | (c) |

**Figure 5.** Fundamental components that demonstrate the development of 5FTH. (**a**) Simplified 5FTH PCB layout ; (**b**) DC/DC converter used in PCB; (**c**) RFM96 applied in PCB.

Taking advantage of the tools mentioned, the first PCB version of the 5FTH module can be assembled. It can be viewed through different angles in Figure 6, with ATMega328p used as the core. In which Figure 6a indicates the sectors distributed within the PCB, where the DC/DC converter is placed as close as possible to the battery and the microcontroller. By this way, in Figure 6b there is a demonstration of the peripherals that need to be arranged on the edge of the board: the DHT11 needs to have an airflow to perform the measurement, the RFM96 communication antenna to not suffer signal blockages, and the flame sensors not be covered by other components. As with HTW development, this step is crucial to check for possible failures between the listed components. Therefore, in the next section, the tests performed during this stage are demonstrated.

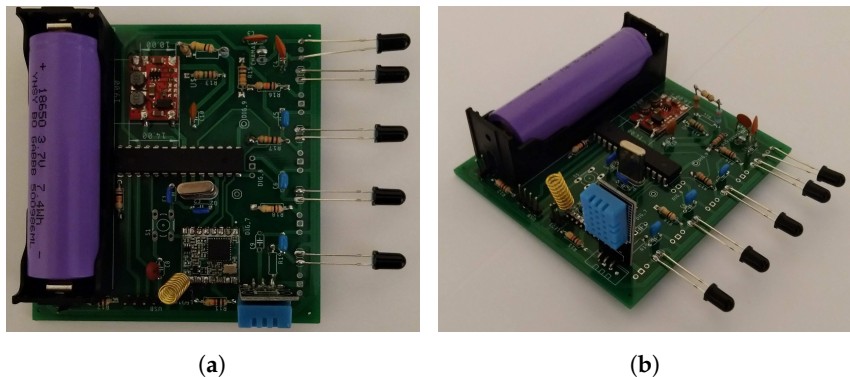

| (a) | (b) |

**Figure 6.** Developed sensor module 5FTH. (**a**) Developed PCB with the components (Top view); (**b**) Developed PCB with the components (Side view).

Figure 7 shows the proposed low-cost prototype module finalized and deposited in a 3D printed box, with dimensions: 100 mm × 40 mm × 90 mm. In Figure 7a, there was a necessity to reorganize some peripherals, such as the arrangement of flame sensors. These were removed from the PCB and connected by wires, then they were fixed to all sides of the box, except the top and back of the 5FTH module. There was no fixation on these faces because the idea is to attach these low-cost modules to the tree trunks (rear face), and also because it is not necessary to point to the sky (upper face). Another peripheral that demanded to be moved is DHT11, as it could not be inside the box. Therefore, a grid was created at the box bottom so that DHT11 could adequately measure the temperature and relative humidity air (Figure 7b).

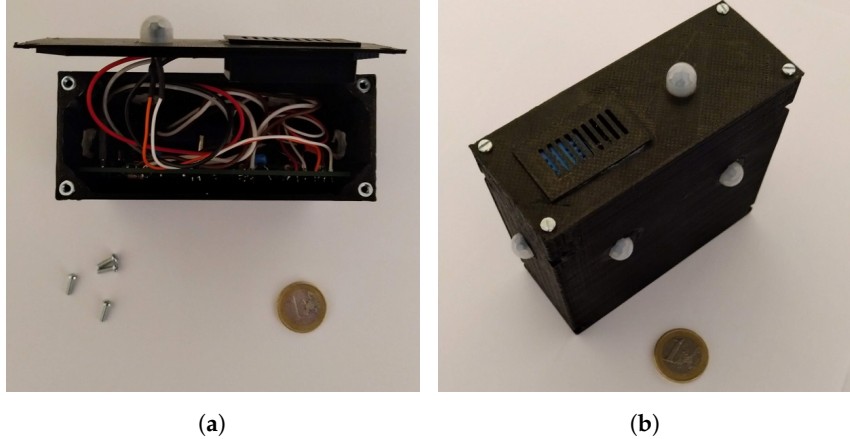

(**a**)  (**b**)

**Figure 7.** Developed sensor module 5FTH in the 3D printed case. (**a**) Top view; (**b**) Isometric view.

### 4.3. Communication System

The developed system uses LoRaWAN as a communication protocol. LoRaWAN defines the communication protocol and system architecture for the network, while the LoRa physical layer enables the long-range communication link. LoRa is a wireless modulation used to create the long-range communication link. Some wireless systems use Frequency Shifting Keying (FSK) modulation because of its efficient modulation for achieving low power. LoRa is based on chirp spread spectrum modulation that maintains low power characteristics as FSK modulation, although it increases the communication range [29].

As the HTW module can have several types of sensors, it was necessary to build a cluster system identical to the Zigbee Cluster Library (ZCL) [30] using the `FPort`, that is part of the `MACPayload` from the LoRaWAN frame format, as "Cluster Identification".

The data are then sent wirelessly via the LoRaWAN protocol to a gateway, Lorix One from Wifx [31], which will send it over an Ethernet connection (available at gateway location through a 4G/LTE modem) to the TheThingsNetwork [32] (TTN) server. In fact, this server is used just for development and testing. Further, a server will be implemented in our research lab facilities. The data on arrival at the TTN are decoded using a Javascript function. The example of one of these functions is in Figure 8. This is a small demonstrative example of the use of two SDI-12 sensors, a 5TE sensor and an ATMOS22, which sends data through port 31 (5TE) and the port 42 (ATMOS22). The decode function shown sends the data to ThingsSpeak [33]. ThingSpeak is an IoT analytics platform service that allows aggregation, visualization and analyses of live data streams using Matlab [34]. ThingSpeak only has eight data fields (the name of this field must be field1 to field8).

```
function Decoder(bytes, port) {
  // Decode an uplink message from a buffer
  // (array) of bytes to an object of fields.
  var decoded = {};
  byte = 0;

  if (port === 31) {
    decoded.permittivity = (((bytes[byte++] << 8) | bytes[byte++]) /
    ↪  100.0).toFixed(3);
    decoded.EC           = (((bytes[byte++] << 8) | bytes[byte++]) /
    ↪  100.0).toFixed(3);
    decoded.soilTemperature  = ((((bytes[byte++] << 8) | bytes[byte++]) / 100.0) -
    ↪  40.0).toFixed(1);
  }

  if (port === 42) {
    decoded.windSpeed = (((bytes[byte++] << 8) | bytes[byte++]) /
    ↪  100.0).toFixed(3);
    decoded.windDirection = (((bytes[byte++] << 8) | bytes[byte++])).toFixed(3);
    decoded.windGust = (((bytes[byte++] << 8) | bytes[byte++]) / 100.0).toFixed(3);
    decoded.temperature  = ((((bytes[byte++] << 8) | bytes[byte++]) / 100.0) -
    ↪  40.0).toFixed(1);
  }

  return {
    field1: decoded.permittivity,
    field2: decoded.EC,
    field3: decoded.soilTemperature,
    field4: decoded.windSpeed,
    field5: decoded.windDirection,
    field6: decoded.windGust,
    field7: decoded.temperature,
  };
}
```

**Figure 8.** Example of one decode function at TheThingsNetwork (TTN) to obtain data from HTW.

On the other hand, during the data collection of the 5FTH module, it was not necessary to use the clustering process applied in HTW. Therefore, the basic settings that the TTN indicated were used to decode the received data. In this sense, the 5FTH modules payload in final message has 14 bytes, distributed in battery level (2 bytes), temperature (1 byte), humidity (1 byte) and five flame sensors' values (2 bytes each). With this 14-byte payload, the minimum interval between sent packets that the TTN server allows to transmit is 60 s. Otherwise, the server may ban the account from applications that use the service for up to 24 h.

## 5. Results

This section will present the results of the developed modules, 5FTH and HTW, regarding the acquisition, transmission, and registration. Firstly, it will be addressed the flame sensors behaviour as a distance dependency. This module is based on a diode of 940 nm flame sensor, that can be used for fire source detection and infrared detection.

### 5.1. Flame Detectors Assay

Before placing the detection prototypes in the forest, some lab tests are necessary to run in a controlled environment. Laboratory tests are essential to ensure that prototypes can acquire data and send them remotely. This avoids possible communication failures, as well as some false alarms. Figure 9 represents the scenario illustration mounted on the laboratory bench, where a candle was

first positioned at 1000 mm for about 300 s. Then, after the first period, this candle was positioned 100 mm closer to the sensors and for the same time. This whole process wasis repeated until the candle reached the tenth point, that is, 100 mm away from the 5FTH sensor module. The following subsections describe the results obtained regarding laboratory tests.

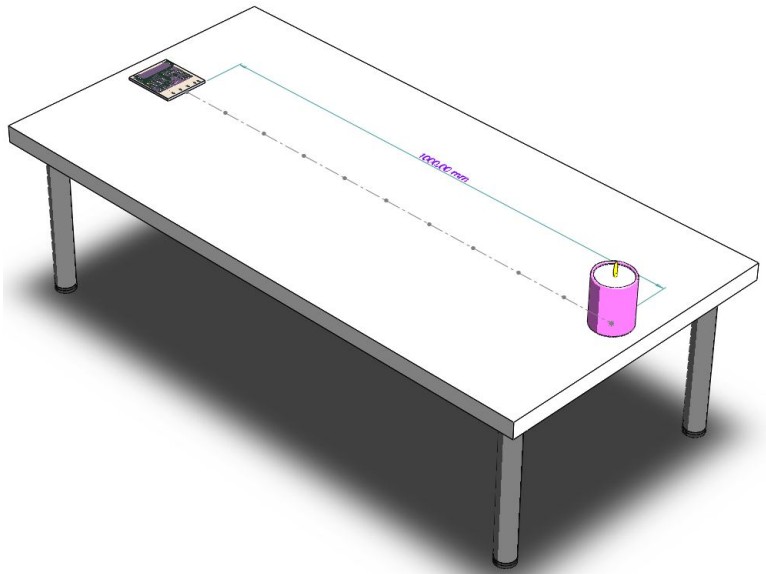

**Figure 9.** Illustration scenario in laboratory bench.

### 5.2. Flame Detection by 5FTH

3D printing and prototype assembly was elaborated and the first evaluation is to analyse if the sensors can work without interruptions. Then, the prototype was turned on for 10 tests collecting data within the laboratory with a 2 s collection interval (using a USB cable).

All data were recorded without interruption, so all sensors were able to collect data at any time of the day and without large peak oscillations. Due to a large amount of data, Figure 10 shows only data collected in a single test.

In the data acquisition graph presented in Figure 10, it is possible to establish the minimums and maximums values of the flame sensors, where values ranged from 0 to 1023 (values from ATMega328P analog input). When the sensor was close to the flame, the values tended to be close to 0. On the other hand, when the sensor moved away from the flame, the values tended to approach 1023. The sensitivity of the sensor varied with distance, intensity and flame volume. This sensitivity can be better studied with standardized tests by fires that follow the ISO 834 standard fire simulation curve, through furnaces that have the EN 1363-1 standard [35]. In this way, it will be possible to establish the relationship between the values measured by the sensors (0-1023) with the flame volume (useful oven volume), the combustion distance (linear or radial) and fire intensity (power supplied by the oven burners).

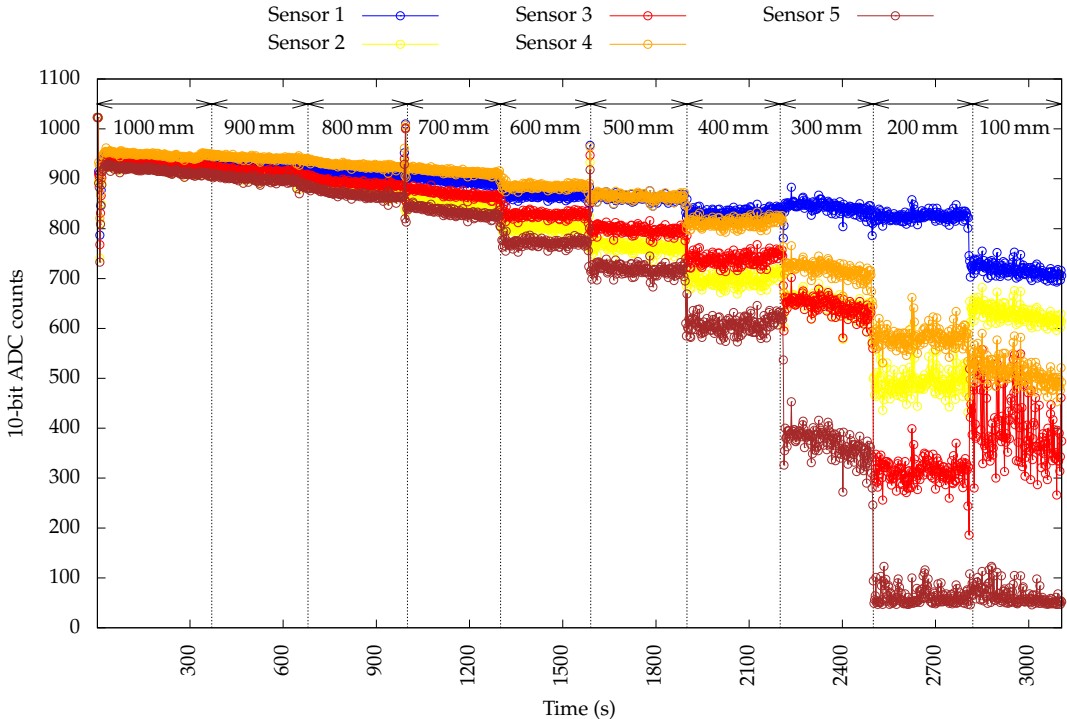

**Figure 10.** Flame sensors data.

By simulating the forest fire ignition with a candle (15 mm), it was possible to detect the presence of fire at about 700 mm to 100 mm. It is also important to note that the differences between the values of the flame sensors may have been caused due to their range of vision, since each one had a radial view range of 24° (means ±12°), as specified by the data sheet. Therefore, the orientation in which they were fixed could also have influenced the acquisition of infrared radiation spectra. The solution that will be implemented on the prototype, is to add Fresnel lenses to adjust the radial view of the sensors.

In short, the test performed with the flame of a candle only demonstrated that the sensors could detect a relatively small flame (15 mm in height) and also that they could provide data in a constant manner (which avoided some signal conditioning).

### 5.3. Acquisition and Registration Data

After the test using a candle on the laboratory bench, it was still necessary to carry out the data collection procedure of the two modules (HTW and 5FTH) through the LoRaWAN network used. In this way, it was possible to analyze the behavior of the devices while processing data using the TTN server, as well as decoding and storing the data. Besides, as the intention was not to compare the performance between the two modules, for this test, the HTW sensor was installed outdoor (forest), and the 5FTH was kept indoors. During the tests, no loss packet report was obtained, and the approximate distance between the Lorix One gateway with the HTW and 5FTH modules was 600 m and 5 m respectively.

A view of the entire system is shown in Figure 11. The two implemented nodes sent the data over the LoRaWAN network to a gateway that was connected over Ethernet to the TTN server. In the TTN server a decoder was implemented that was related to the developed library (i.e., the FPort will identify the type of decoder to be used—e.g., see Figure 8). After decoding the data were sent to the ThingSpeak platform for visualization and possible analysis and aggregation.

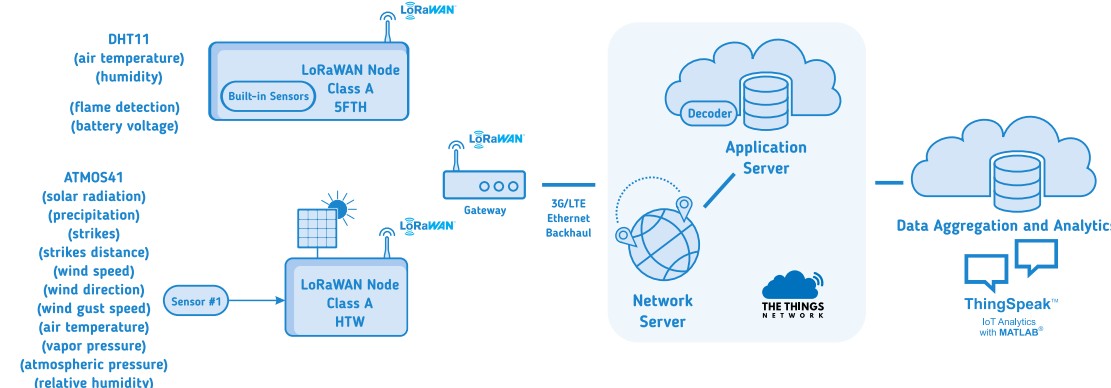

**Figure 11.** Overview of the implemented system.

### 5.3.1. HTW Communication

In the case of the node HTW with ATMOS41, all-in-one weather station, only four of the possible 12 measured parameters are shown (Figure 12). Although the 12 parameters were sent, only eight could be sent to ThingSpeak. The presentation of solar radiation, air temperature (Figure 12a), barometric pressure, and precipitation (Figure 12b) was opted for.

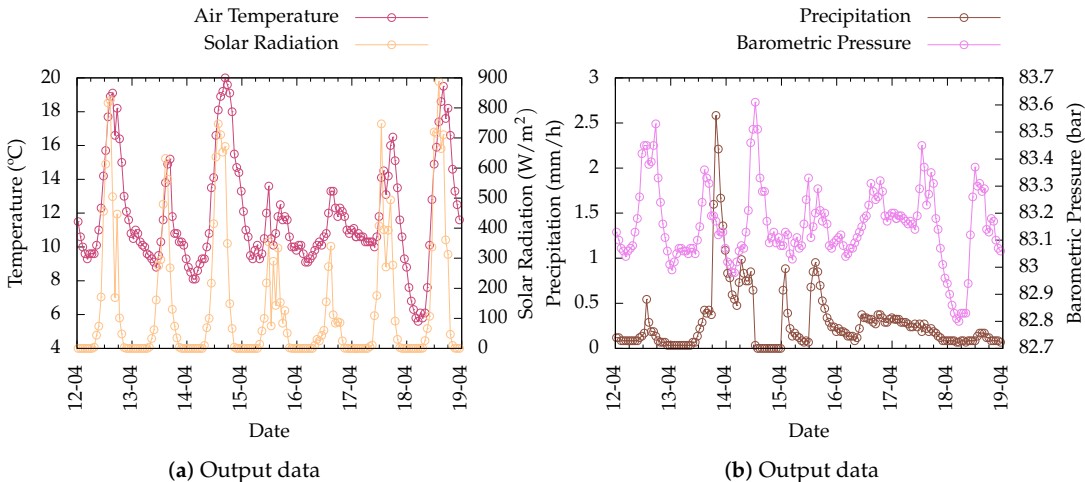

(**a**) Output data  (**b**) Output data

**Figure 12.** Data from LoRaWAN node with ATMOS41 all-in-one weather station.

This module kept its battery at full charge continually for at least six months; that is, no data were recorded below 3.7 V during that period even when the transmission period rate was decreased. It demonstrates that the solar panel chosen for this module validates its autonomy, being able to be installed in remote regions or awkward to access for maintenance (as long as the communication signal reached.

### 5.3.2. 5FTH Communication

The more frequently a transmission rate is set, the higher the battery's energy discharge (the device consumes more times in a shorter period). However, it is not possible to choose any transmission rate, due to the transmission policies of the TTN server (mentioned in Section 4.3). Concerning this point, in the case of the transmission made by 5FTH, the data were collected with 60 s in the transmission rate to intensify the battery discharge and without sleep mode activated. In this sense, it is possible to analyze not only the data transmission but also if the battery measurement had the expected behavior. Figure 13 shows the data collected from the DHT11 sensor, that is, the temperature and relative humidity air.

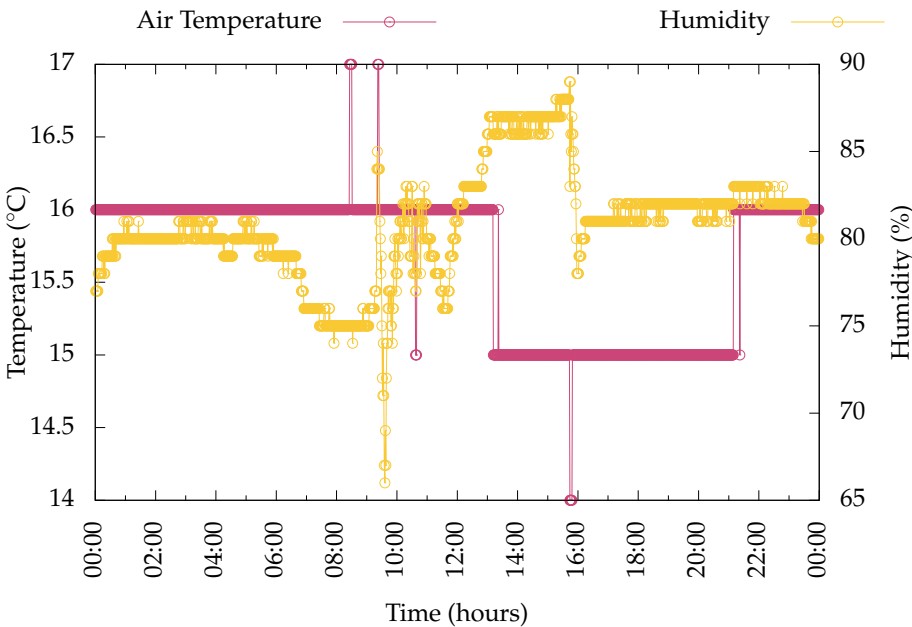

**Figure 13.** Air temperature and humidity data from DHT11 sensor at node 5FTH.

In the first five hours of this test, a candle was left between 40 mm to 90 mm from the 5FTH, and then removed until 24 h. As the objective was only to test the communication, exclusively the data from Sensor 5 are shown in the Figure 14, together with the data transmitted from the battery voltage.

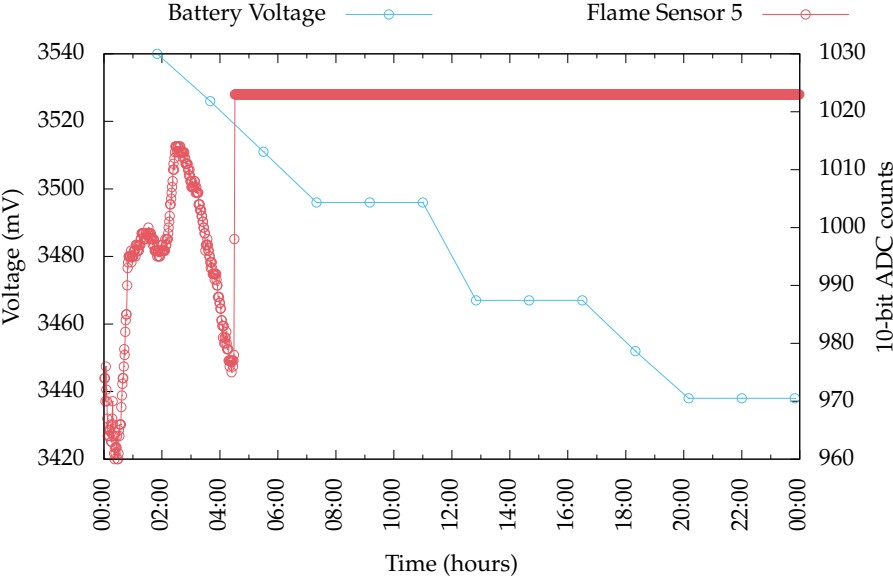

**Figure 14.** Battery voltage and flame sensor data from node 5FTH.

With these configurations, the module remained to send data via LoRaWAN using the TTN server for approximately seven days. It would be the worst configuration scenario for this module since all energy optimizations were ignored in this test. Possible strategies can still be mixed in a study only for energy optimization of the 5FTH module, such as the use of sleep mode with the lowest possible transmission rate. However, the test demonstrates that the module was capable of continually sending data over a complete battery cycle, ranging from maximum charge to minimum charge.

## 6. Conclusions and Future Work

The Forest Alert Monitoring System (SAFe) project proposes to develop innovative operations to minimize the alert time of forest fires ignitions. This project will contribute to the actual surveillance systems, enhancing the firefighters and civil protection with more details and real-time information. As a base of this project, the acquisition and communication modules, that will be spread in the forest, will gather information about several relevant data for efficient characterization of existing forest conditions. This work presents an approach for the wireless sensor network based on two types of sensors modules: 5TFTH and HTW. These sensor modules work in a collaborative way in order to obtain different type of data from the forest. The preliminary tests indicate that the proposed and developed modules operate as expected, both in the acquisition, transmission, and data recording. This recorded data can be used in the future, resorting to artificial intelligence, pattern recognition, and cluster algorithms that will give indications to alert and predict fire ignitions. As next steps, the team will use an optimization algorithm based on genetic algorithm to identify the number of sensors modules and their optimal position in order to optimize the collaborative communication between them and guarantee the surveillance of the maximum forest area. The improvement of the developed modules with industrial-grade components and waterproof enclosure can also be point to be addressed.

**Author Contributions:** The contributions of the authors of this work are pointed as follows: Conceptualization, A.I.P., A.V. and J.L.; Methodology, A.I.P. and T.B.; Software, A.V. and T.B.; Validation, A.V and T.B.; Investigation, A.I.P., A.V., J.L. and T.B.; Data Curation, A.V. and T.B.; Writing—Original Draft Preparation, A.I.P., A.V., J.L. and T.B.; Writing—Review and Editing, A.I.P., A.V., J.L. and T.B.; Visualization, T.B. and A.V.; Supervision, A.I.P., A.V. and J.L.; Project Administration, A.I.P. and J.L. All authors have read and agreed to the published version of the manuscript.

**Funding:** This work is financed by SAFe Project through PROMOVE—Fundação La Caixa.

**Acknowledgments:** Authors would like to thank the the firefighters and civil protection entity. This work has been supported by FCT—Fundação para a Ciência e Tecnologia within the Projects Scope UIDB/05757/2020.

**Conflicts of Interest:** The authors declare no conflict of interest.

## Abbreviations

The following abbreviations are used in this manuscript:

| | |
|---|---|
| ADC | Analog-to-Digital Converter |
| DC/DC | Direct Current to Direct Current |
| FSK | Frequency Shifting Keying |
| I2C or I$^2$C | Inter-Integrated Circuit |
| IoT | Internet of Things |
| LPWAN | Low Power Wide Area Network |
| LoRa | Long Range protocol developed by Semtech–physical layer |
| LoRaWAN | Long Rang Wide Area Network–networking layers |
| NaaS | Network and as a Service |
| PCB | Printed Circuit Board |
| SAFe | Sistema de Monitorização de Alerta Florestal (Forest Alert Monitoring System) |
| SPI | Serial Digital Interface |
| SPI-12 | Serial Digital Interface at 1200 baud |
| TTN | The Things Network |
| UART | Universal Asynchronous Receiver-Transmitter |
| WSN | Wireless Sensor Network |

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
