# Peer review of "Wireless Sensor Network for Ignitions Detection: An IoT approach"

_electronics, doi:10.3390/electronics9060893_

Round 1

Reviewer 1 Report

I consider that the presented work is interesting but seems very immature at this stage and needs an extra effort to be publishable.

There are commercial solutions covering fire detection that follow, more or less, the same approach.

Some comments and weakness below:

  • line 52: something missing about Sigfox, rewrite
  • line 57: LoRaWAN is not a public network offered by a network operator. This wor use LoRaWAN so you should correct that. For example, it is very interesting about LoRaWAN that you can deploy your own infrastructure.
  • line 69: this election is not sustained by that conclusion. Take into consideration that LoRaWAN require to deploy your own gateway, so you incur an extra cost. For example, this argument is not valid if you consider Sigfox in the mix. Justify in another way, for example, "there is no Sigfox coverage ..."
  • line 104: "??? WireSesnor Modules??? Is that correct?
  • Section 4. Some of the components are not adequate for industrial range temperature, e.g CubeCell, DHt11, ... ) taking into consideration that this must operate in a forest, it is fundamental to elect industrial-grade electronic components.
  • Also, this should be applied to the housing that, at least, will suffer water condensation (as I can check visually in figure 7. LiPo battery has problems of stability in low temperatures, please indicate the model.
  • Could you please provide more information about the "flame detector". Perhaps this information is in reference [19], but this reference is placed in a diagram of the whole system.
  • Eliminate figure 8 if you do not describe the protocol.
  • Line 219. Wi-Fi: I assume that this is only for lab experiments. Please add information about the elected gateway.
  • TheThingsNetwork must not be considered for professional deployments, it is only for amateur and open projects because the network servers tend to fail. TTN "TheThingsIndustries" commercial solution Loriot, senet or similars should be considered in a real scenario.
  • line 220: You don't use Java in the decoder of TTN. This is javascript
  • Code provided in figure 9 has errors, e.g "(port === 31)"
  • Results section lacks the most important aspect of a wireless sensor, that is, the communication behavior of LoRaWAN in a forest. This is a "must have" or this work can see as a preliminary work.

  •  
  •  

Reviewer 2 Report

In this paper, a LoRaWAN based wireless sensor network is built for forest ignition detection. Two models, 5FTH and HTW, are also utilized in the proposed system. The system architecture is clear and the experiment results are also comprehensive. Besides, this paper is well-organized and easy to read. Overall, this is a worthy paper to be published. There are two suggestions for the authors. Firstly, the contributions of this paper should be clearly described in the first section. Secondly, the abstract should be rewritten. Simply describe the method and major result in the abstract so that the readers can easily get the concept of this paper.

Reviewer 3 Report

Thank you for the submission of this article for review, unfortunately it needs a significant amount of work before it can be published.

The quality of the language used throughout the report although particularly in the introduction make it very hard to understand, so a very thorough proof read and edit is needed.  It should also be noted that "Ignitions" is not a word.

Line 3: please give examples of the variables that you intend to measure.

Abstract: This should summarise the entire paper and include some details as to the contributions.  Currently this is too short / high-level to explain the outcomes of the paper.

Line 21: I was expecting the AI to be described in the paper, it is only later on that you scope it to just the hardware, this should be made clear early in the introduction.

Line 27 & 29: Do 5FTH & HTW mean anything?

Lines 49 - 51: This jumps straight from WSN into IoT without explaining the differences / similarities between them.

Line69: Please justify your choice of LoRaWAN in more detail than just a sentence.

Figure 1: Figures should be able to stand alone without referring to the text.  Please explain the meaning of the numbers in the caption.

Line 112: The sensor nodes have internet access? (4) This isn't mentioned anywhere else in the text?

Line 179: Why a 60 second sample period? Is this standard in the literature if so quote it, otherwise please justify.

How were each of the sensors (such as the DHT11) chosen?

Section 4.3 The details of FSK / CSS modulation and the MAC Payload are part of the LoRaWAN standard and so should be under section 2 as they are related work.  The specific packing for this project can be here, but the complete stack shouldn't be.

Line 223: Why ThingsSpeak?

Line 239: Is the max detection range of the sensor 1m, if so what density of sensors will you need to get meaningful coverage of the forest? This should be discussed in the paper.

Figure 11: What does this picture show? For those not used to working with flame sensors what does it mean? Is a high value flame detected or low?

Line 283: What distance was it left at? Measurements should not be described as "about" in formal literature.

For the communication tests what was the range between node and gateway? What is the gateway? What is the estimated range that will be used for the deployment? Was any packet loss observed? Has LoRaWAN been tested to work in the forest?

Figure 15: Have any calculations been done as to battery life?  Will the solar panel get enough energy for the system to last indefinitely?

This work has the potential to be a great paper, but unfortunately at the moment this paper needs significant work.  I hope the comments above will help to improve this paper.

Reviewer 4 Report

Dear author,

Those paragraphs are detected plagiarism, please rewrite by your own language:

Line 36-37: Rewrite this sentence to prevent plagiarism.

Line 65-68:  Rewrite this sentence to prevent plagiarism.

Line 80-82:  Rewrite this sentence to prevent plagiarism.

Line 117-118:  Rewrite this sentence to prevent plagiarism.

Line 207-213:  Rewrite this sentence to prevent plagiarism.

Line 244-249:  Rewrite this sentence to prevent plagiarism.

Line 250-255:  Rewrite this sentence to prevent plagiarism.

Round 2

Reviewer 1 Report

I consider that the requested enhancements has been achieved or has been adequately explained.

Reviewer 4 Report

The author did rewrite and improved a lot. The paper has good construction and met the requirements of this journal. 

THank you.